# Physiological and Transcriptomic Analysis Provide Insight into Low Temperature Enhancing Hypericin Biosynthesis in *Hypericum perforatum*

**DOI:** 10.3390/molecules26082294

**Published:** 2021-04-15

**Authors:** Hongyan Su, Jie Li, Sijin Chen, Ping Sun, Hua Xing, Delong Yang, Xiaona Zhang, Mengfei Li, Jianhe Wei

**Affiliations:** 1Key Lab of Arid Land Crop Science/College of Life Science and Technology, Gansu Agricultural University, Lanzhou 730070, China; Shy922322@163.com (H.S.); lijie9654@163.com (J.L.); chenjj@gsau.edu.cn (S.C.); sunp@gsau.edu.cn (P.S.); xingh@gsau.edu.cn (H.X.); yangdl@gsau.edu.cn (D.Y.); 2Gansu Herbal Medicine Planting Co., Ltd., Lanzhou 730000, China; zhangxiaona@gansubaicao.com; 3Institute of Medicinal Plant Development, Chinese Academy of Medical Sciences & Peking Union Medical College, Beijing 100193, China

**Keywords:** *Hypericum perforatum*, hypericin biosynthesis, low temperature, physiological characteristic, transcriptomic analysis, gene expression

## Abstract

Hypericin (Hyp), well-known as an antidepressant, is mainly extracted from *Hypericum perforatum*. Although Hyp accumulation and biomass are greater at lower compared to higher temperature, the regulation mechanism has not been reported. Here, the physiological characteristics and transcriptome of *H. perforatum* grown at 15 and 22 °C were determined and analyzed by HPLC and de novo sequencing. The results showed that the stomatal density and opening percentages were 1.1- and 1.4-fold more, and the Hyp content was 4.5-fold greater at 15 °C compared to 22 °C. A total of 1584 differentially expressed genes (DEGs) were observed at 15 versus 22 °C, with 749 characterized genes, 421 upregulated (UR) and 328 downregulated (DR). Based on biological functions, 150 genes were associated with Hyp biosynthesis, plant growth and the stress response, including photosynthesis, carbohydrate metabolism, fatty acids metabolism, cytochrome P450 (CYPs), morpho-physiological traits, heat shock proteins (HSPs), cold-responsive proteins (CRPs) and transcription factors (TFs). The differential expression levels of the master genes were confirmed by qRT-PCR and almost consistent with their Reads Per kb per Million (RPKM) values. This physiological and transcriptomic analyses provided insight into the regulation mechanisms of low temperature enhancing Hyp biosynthesis in *H. perforatum*.

## 1. Introduction

*Hypericum perforatum* L. (St John’s wort) is a perennial herbaceous plant native to Europe, Northern Africa, Northern America and Asia [1,2]. It has been used in folk medicine as a soothing, antiphlogistic and urogenital agent, as well as in the treatment of traumas, burns, scabs and hemorrhoids since Greek and Roman times [3]. Today, it is mainly used for antidepression agents [4,5,6], which largely depend on the naphthodianthrone compound Hyp [3,7]. Currently, Hyp is mainly extracted from *H. perforatum* [8]. In order to improve the Hyp production, many approaches that have been investigated include dissecting the Hyp biosynthetic pathway [9,10,11], screening germplasm resources [12,13,14], evaluating harvest stages [8,12,15], finding out the optimization of environmental factors such as nutrients [16], drought stress [17], UV-B radiation [18,19,20], light [21] and temperature [12,20,22,23].

Hyp biosynthesis is started from glucose to acetyl- and malonyl-coenzyme A (CoA) in the cells of green tissues and from acetyl- and malonyl-CoA to Hyp in the cells of dark glands (Figure 1). Briefly, in green tissue, photosynthesis firstly produces glucose, and pyruvic acid is produced through glycolysis. Acetyl-CoA can be formed through pyruvate dehydrogenase. Malonyl-CoA can be derived from acetyl-CoA by acetyl-CoA carboxylase. In addition, acetyl-CoA can also be recycled as the result of the fatty acid metabolism. In the dark gland, one molecule of acetyl-CoA and seven molecules of malonyl-CoA form an octa-β-ketoacyl chain by polyketide synthase (PKS) or octaketide synthase (OKS); emodin anthrone was formed after several steps, including aldolic condensation, thioesterase (TER), decarboxylic and dehydration reactions. Emodin dianthrone can be generated through two pathways, including the oxidation of emodin bianthrone, which is formed by the free radical coupling of emodin anthrone, and the synthesis of emodin anthrone and emodin by the phenoloxidative coupling protein (POCP). Then, Hyp is subsequently yielded via the catalysis of the berberine bridge enzyme (BBE) [9,10,11,22,24,25,26].

Experimental evidence shows that a low temperature improves Hyp production. Specifically, Brunáková et al. [27] reported that a −4 °C treatment improved the Hyp content on a dry weight (DW) basis. Tavakoli et al. [20] found that the Hyp content on a DW basis in a root suspension culture reached the highest amount at 4 °C than at 8, 16 and 25 °C. Velada et al. [28] determined that the genes (phenolic oxidative coupling protein (*Hyp-1*), *PKS1* and *PKS2*) involved in Hyp synthesis were differentially expressed at different temperatures. Our previous studies found that 15 °C could enhance aerial parts of a biomass and Hyp contents of dry weight (DW) on a per plant basis, as well as mRNA expressions of the *hyp-1* and *PKS* genes, compared to 22 °C [22]. For the molecular function of *hyp-1* gene, Bais et al. [29] found that a conversion of emodin to Hyp was solely catalyzed by the *hyp-1* gene, while Košuth et al. [30] reported that the *hyp-1* gene is not a limiting factor for Hyp biosynthesis in the genus *Hypericum* or that additional factors are necessary for completion.

To date, the mechanism of low temperature regulating plant growth and Hyp biosynthesis in *H. perforatum* is still limited. With the aim of unraveling the genes involved in growth enhancement and Hyp accumulation upon low temperature induction, the seedlings grown at 15 and 22 °C were examined by RNA-seq, the mRNA expression levels were confirmed by qRT-PCR and the Hyp content was quantified by HPLC.

## 2. Results

### 2.1. Effect of Low Temperature on Biomass, Dark Glands, Leaf Stomata and Hyp Accumulation

A significant increase of the DW of whole plants was observed at 15 °C than 22 °C (Figure 2A,C). Although 1.1-fold more of the number of foliar dark glands was observed at 15 °C than 22 °C, there was no significant difference between them (Figure 2B,D). Four point six and 4.5-fold increases of the Hyp content, on a DW and per plant basis, were observed at 15 °C compared to 22 °C, respectively (Figure 2E,F). Additionally, there were significant differences in the leaf stomata, with the stomatal density and opening percentage 1.1- and 1.4-fold more (Figure 3A,C and Table 1 and a more rounded shape with the ratio of length/width (2.7) at 15 °C compared to 22 °C (Figure 3B,D and Table 1).

### 2.2. Global Gene Analysis

To dissect the mechanisms of low temperatures enhancing plant growth and Hyp accumulation, RNA-seq for plants grown at 15 and 22 °C was performed. Although robust data were generated (Appendix A), 39.4 and 45.5 million high-quality reads were collected after data filtering, and 2.5 and 2.9 million multiple reads were mapped from the 15 and 22 °C plants, respectively, from the 44,776 compiled genes and annotated against the databases (Appendix A). The correlation between 15 and 22 °C was assessed by a principal component analysis (PCA) with R packages (Appendix A). A total of 1584 differentially expressed genes (DEGs) were identified at 15 versus 22 °C (Appendix A). Of these 1584 DEGs, 768 genes were identified to match with the databases (Figure 4B). Among the 768 genes, 749 genes with known functions were partitioned into 421 upregulated (UR) and 328 downregulated (DR) (Figure 4C,D).

### 2.3. Biological Category of DEGs

Based on biological functions, the 749 genes were divided into 11 categories: photosynthesis/energy (25), primary metabolism (85), secondary metabolism (62), cell morphogenesis (60), bio-signaling (93), hormone biosynthesis (23), polynucleotide biosynthesis (33), transcription factor (112), translation (58), transport (104) and stress response (94) (Figure 4D). Based on the pathways for the control of Hyp biosynthesis, plant growth and stress response [10,20], finally, 150 DEGs (90 UR and 60 DR) were dug out as master genes involved in photosynthesis (12), carbohydrate metabolism (13), fatty acid metabolism (15), cytochrome P450 (21), morphophysiological traits (15), heat shock protein (15), cold-responsive protein (3) and transcription factors (56). The differential expression levels of the 150 genes based on Reads Per kb per Million (RPKM) values are shown in a heat map (Figure 5 and Appendix A).

### 2.4. DEGs Involved in Hyp Biosynthesis

#### 2.4.1. DEGs Involved in Photosynthesis

As is mentioned above in the Hyp biosynthetic pathway (Figure 1), Hyp biosynthesis is initiated from photosynthesis producing glucose in the cells of green tissues. Here, 12 genes (9 UR and 3 DR) that are involved in photosynthesis include chlorophyll a-b binding proteins (CABs) (CAB1, CAB1B, CAB3, CAB3C, CAB96 and CAB); early light-induced proteins (ELI_PEA and ELIP2); light-harvesting complex-like protein (OHP2) and ribulose bisphosphate carboxylases (RBCS-C, RCA and RBCS-8B) (Figure 5 and Appendix A). The differential UR of the selected six genes encoding the CABs was confirmed by qRT-PCR. Their RELs are consistent with RPKM values, with 1.1-, 2.9-, 2.5-, 1.7-, 3.9- and 3.6-fold UR of CAB1, CAB1B, CAB3, CAB3C, CAB96 and CAB, respectively, at 15 versus 22 °C (Figure 6A).

#### 2.4.2. DEGs Involved in Carbohydrate Metabolism

Glucose accumulation is indispensable for the upstream of Hyp biosynthesis (Figure 1). Here, 13 genes (10 UR and 3 DR) that were involved in carbohydrate metabolism included the carbon reaction (CA, CA1, LDHA and PGAM) and glucose metabolism (AGLU2, BGLU11, GINT1, GAD, GAPC, NonBGLU, TOGT1, GAE6 and USP1) (Figure 5 and Appendix A). Differential UR of the selected seven genes was confirmed, and their RELs were consistent with the RPKM values, with 1.3-, 1.8-, 2.6-, 4.3-, 5.4-, 1.2- and 1.2-fold UR of CA, CA1, LDHA, PGAM, BGLU11, GAD and GAPC, respectively, at 15 versus 22 °C (Figure 6B).

#### 2.4.3. DEGs Involved in Fatty Acids Metabolism

Another important way to generate acetyl-CoA is fatty acid metabolism. Here, 15 genes (12 UR and 3 DR) that are involved in fatty acid metabolism included FAD3, YLR118C, abhd17c, HOS3, FAR3, MGD3, At1g06800, At4g16820, PLDZETA1, PLD1, SQD2, LIP2, BEAT, BPS and PED1 (Figure 5 and Appendix A). Differential UR of the selected four genes was confirmed, and their RELs were consistent with the RPKM values, with 1.1, 2.4-, 2.6- and 4.1-fold UR of PLDZETA1, BEAT, BPS and PED1, respectively, at 15 versus 22 °C (Figure 6C).

#### 2.4.4. Differentially Expressed Cytochrome P450 (CYPs)

In this study, the 21 genes (17 UR and 4 DR) that encoded CYPs included CYP71A6, CYP71A9, CYP78A5, CYP82A1, CYP82A3, CYP89A9, CYP86B1, CYP90B1, CYP94B1, CYP96A15, CYP714C2, CYP81D1, CYP81E8, CYP716B1, CYP736A12, CYP749A22, CYP82D47, CYP75B2, CYP93B1, CYP81E9 and CYP81E1 (Figure 5 and Appendix A). Differential UR of the selected two genes was confirmed, and their RELs were consistent with RPKM values, with 4.5- and 1.5-fold UR of CYP75B2 and CYP81E1, respectively, at 15 versus 22 °C (Figure 6C).

### 2.5. DEGs Involved in Morpho-Physiological Traits

In this study, 15 genes that directly participated in the morphophysiological traits included leaf and stem development (ZOG1, DCR and CYP96A15); root development (MIZ1 and GEM); flower development (CET1, FPF1, FTIP1, HD3B and SWC6) and cell differentiation (SBT1.9, SBT3.18, SBTI1.1, EMS1 and MSP1) (Figure 5 and Appendix A). Differential expression of the selected six genes was confirmed, and their RELs were consistent with the RPKM values, with the 1.8-, 2.4-, 0.05-, 0.1-, 2.2- and 0.04-fold differential expression of CYP96A15, EMS1, GEM, MIZ1, SBTI1.1 and ZOG1, respectively, at 15 versus 22 °C (Figure 6D).

### 2.6. DEGs Involved in Stress Response

#### 2.6.1. DEGs Involved in Heat Shock Proteins (HSPs) and Cold-Responsive Proteins (CRPs)

Since *H. perforatum* seedlings were exposed to 15 and 22 °C in this study, 18 genes that were observed to be differentially expressed included 15 DR HSPs (HSP15.7, HSP17.3-B, HSP17.4B, HSP17.9-D, HSP18.2, HSP22.0, HSP70-8, HSP70, HSC-I, HSC-2, HSF30, HSP81-1, HSP83A, HSP21 and HSA32) and three UR CRPs (BAP1, COR413IM1 and CRPK1) (Figure 5 and Appendix A). Differential regulation of the selected six genes was confirmed, and their RELs were consistent with the RPKM values, with 0.3-, 0.98- and 0.5-fold DR of HSP70, HSC-I and HSC-2 and 2.9-, 11.3- and 1.6-fold UR of BAP1, COR413IM1 and CRPK1, respectively, at 15 versus 22 °C (Figure 6E). The DR of HSP genes and UR of CRP genes can be an adaptive mechanism of *H. perforatum* in response to low temperatures.

#### 2.6.2. Differentially Expressed Transcription Factors (TFs)

In this study, 56 TFs that were differentially expressed included MYB TFs (PHL5, MYB2, MYB3, MYB4, MYB14, MYB17, MYB48, MYB52, MYB62, MYB73, MYB78 and MYB108); MADS-box TFs (AGL80, SVP, CMB1 and MADS50); bHLH TFs (BHLH62, BHLH92, BHLH96, BHLH154 and BHLH162); TCP TFs (TCP1, TCP12 and TCP20); bZIP TFs (BZIP44, BZIP61 and TGA10); WRKY TFs (WRKY7, WRKY40, WRKY41, WRKY53, WRKY72, WRKY6, WRKY28, WRKY55 and WRKY24); ethylene-responsive TFs (At1g50680, At1g51120, ERF2, ERF2, ERF5, ABR1, ERF003, ERF013, ERF017, ERF020, ERF071, ERF095 and ERF109) and NAC domain-containing proteins (NAC014, NAC043, NAC048, NAC071, NAC087, NAC090 and NAC100) (Figure 5 and Appendix A). Differential UR of the selected six genes was confirmed, and their RELs were consistent with the RPKM values, with 1.4-, 4.4-, 2.3-, 2.1-, 2.2- and 1.3-fold UR of ERF5, ERF020, MYB4, MYB14, NAC071 and WRKY24, respectively, at 15 versus 22 °C (Figure 6F).

## 3. Discussion

While low temperatures improving the plant growth and Hyp accumulation by stimulating related gene expressions has been observed in *H. perforatum* [20,22,27,28], the mechanism responsible for temperature-dependent growth and Hyp biosynthesis has not been dissected. Here, it is shown that there is a greater plant biomass and Hyp accumulation, more leaf dark glands, stomatal density and rounded stomatal shape at 15 compared to 22 °C. By treating seedlings at 15 and 22 °C, ca. 1600 genes were differentially expressed, with over 56% of the 749 characterized genes being UR at 15 °C compared to 22 °C (Figure 4). By grouping genes based on biological functions, 150 genes were associated with Hyp biosynthesis, plant growth and stress responses, including photosynthesis, carbohydrate metabolism, fatty acid metabolism, CYPs, morphophysiological traits, HSPs, CRPs and TFs (Figure 5).

Experimental evidence shows that *H. perforatum* plants can adapt to low temperatures by increasing the biomass and morphogenesis (e.g., leaf area, number of stem and its internode and root), and low temperatures can improve Hyp biosynthesis by increasing the number of dark glands and related gene expressions [10,12,20,22,27,28]. The current study exhibited greater biomass by increasing the stomatal density and opening percentages, as well as a more rounded stomatal shape with the ratio of length/width and greater Hyp content via the overexpression of 61 genes involved in Hyp biosynthesis (Figure 2 and Figure 3 and Table 1).

As is shown in Figure 1, the Hyp biosynthetic pathway depends on photosynthesis and is involved in carbohydrate metabolism and fatty acid metabolism in the upstream steps, as well as PKS, TER, POCR and CYPs. For the photosynthesis, the CABs are the most abundant protein complex on the thylakoid membrane and are light receptors that capture and deliver excitation energy to photosystems [31]. The UR of CAB genes that are involved in photosynthesis provides energy and fixation and a reduction of CO_2_ into carbohydrate [32].

For the carbohydrate metabolism, the carbonic anhydrases (CAs) catalyze the reversible interconversion between carbon dioxide and bicarbonate [33]. The L-lactate dehydrogenase A (LDHA) synthesizes lactate from pyruvate in glycolysis [34]. The phosphoglycerate mutase-like protein (PGAM) catalyzes the reversible interconversion between 3-phosphoglycerate and 2-phosphoglycerate [35]. The beta-glucosidase (BGLU11) hydrolyzes the terminal, nonreducing beta-d-glucosyl residues to generate beta-d-glucose [36]. The glutamate decarboxylase (GAD) catalyzes the conversion of glutamate to CO_2_ and g-aminobutyrate [37]. The glyceraldehyde-3-phosphate dehydrogenase (GAPC) catalyzes D-glyceraldehyde 3-phosphate to 3-phospho-d-glyceroyl phosphate [38]. The UR of genes that are involved in carbohydrate metabolism enhances the accumulation of pyruvic acid by glycolysis pathways, which provide substrates for the biosynthesis of acetyl-CoA.

For the fatty acid metabolism, for example, the phospholipase D zeta 1 (PLDZETA1) is involved in the lipid catabolic process by the hydrolysis of glycerol-phospholipids to produce phosphatidic acids [39]. The acetyl-CoA–benzyl alcohol acetyltransferase (BEAT) is involved in the biosynthesis of benzyl acetate [40]. The 2,4,6-trihydroxybenzophenone synthase (BPS) is involved in the malonyl-CoA metabolic process [41]. 3-ketoacyl-CoA thiolase 2 (PED1) is involved in long chain fatty acid beta-oxidation [42]. The fatty acid metabolic process will provide abundant acetyl-CoA, which can combine with its product malonyl-CoA to produce an octa-β-ketoacyl chain.

Extensive studies have demonstrated that CYPs can catalyze several reactions, such as hydroxylation, dealkylation and oxidation reactions. Studies have demonstrated that the CYPs are a large and diverse family involved in secondary metabolism [43,44,45,46,47]. During the downstream of the Hyp biosynthetic pathway in dark glands, most of the steps are catalyzed by the oxidation, reduction, dehydration and decarboxylic reactions, as well as free radical coupling and electron-paired donors (Figure 1). For example, flavonoid 3′-monooxygenase (CYP75B2) catalyzes the 3′-hydroxylation of flavonoid B-ring to a 3′, 4′-hydroxylated state [48], isoflavone 2′-hydroxylase (CYP81E1) catalyzes the hydroxylation of isoflavones to 2′-hydroxyisoflavones, daidzein to 2′-hydroxydaidzein and formononetin to 2′-hydroxyformononetin [49]. The UR of genes that encode CYPs may play critical roles in Hyp biosynthesis in the downstream pathway.

However, the genes involved in downstream steps (e.g., PKS, OKS, TER, POCP and BBE) were not examined to be differentially expressed, except for the CYPs class (Figure 1). While several studies were devoted to dissecting the Hyp biosynthetic pathway [9,10,11,22,24,25,26], limited genes involved in the downstream steps were submitted to the public databases (e.g., NR, Swiss-Prot, KEGG, KOG and GO), which led to differentially expressed contigs that could not be mapped and were listed as unidentified or uncharacterized genes in this study.

Adaptation to environmental stresses results from integrated events occurring at all levels of organization, not only from the anatomical, cellular, biochemical and molecular levels but, also, from the morphological level [32]. For example, the alkane hydroxylase MAH1 (CYP96A15) is involved in wax biosynthesis during plant growth [50]. The Leucine-rich repeat receptor protein kinase (EMS1) can mediate signals that control the fate of reproductive cells and their contiguous somatic cells [51]. The GLABRA2 expression modulator (GEM) is involved in the spatial control of cell division and differentiation of root epidermal cells [52]. MIZU-KUSSEI 1 (MIZ1) plays an important role in lateral root development [53]. Subtilisin-like serine proteases (SBTI1.1) can promote plant cell differentiation, organogenesis and somatic embryogenesis, as well as cell proliferation [54]. Zeatin O-glucosyltransferase (ZOG1) catalyzes the formation of O-xylosylzeatin in plant development [55]. The differential expression of the 15 genes involved in the morphophysiological traits may be an adaptation to low temperatures.

There is considerable evidence that HSPs are essential for plant survival under abiotic stresses [56,57]. The expression of HSP genes is observed to be induced by high temperatures and inhibited by low temperatures [58,59]. For example, CRPs are induced to enable plants to survive from chilling and freezing stress [60]. Three CRPs that were identified in this study—for example, BON1-associated protein 1 (BAP1), cold-regulated 413 inner membrane protein 1 (COR413IM1) and cold-responsive protein kinase 1 (CRPK1)—were observed to be differentially expressed in response to cold stress [61,62,63].

TFs can regulate plant growth and metabolism in response to abiotic stresses [64]. For example, ethylene-responsive transcription factors (ERFs) are involved in regulating gene expression and participating in signal transduction pathways [65,66]. MYB transcription factors play regulatory roles in developmental processes and defense responses, the MYB4 is involved in scavenging reactive oxygen species during cold stress [67] and MYB14 induces the expression of *CBF* genes to increase their freezing tolerance [68]. The NAC domain-containing protein family may act downstream of N-rich protein (NRP)-A or NRP-B in the integration of endoplasmic reticulum stress and osmotic stress cell death signals [69]. WRKY domain transcription factors play an important role in plant responses to both biotic and abiotic stress by regulating ABA signaling [70].

Based on the above functional analysis, the schematic representation of the proposed pathways of genes for regulating Hyp biosynthesis, morphophysiological traits and the stress response are shown in Figure 7. Briefly, low temperatures induce the differential expression of genes involved in photosynthesis, carbohydrate metabolism and fatty acid metabolism, which improves the accumulation of primary metabolites, as well as the metabolic processes of carbohydrate and fatty acids. Hyp biosynthesis is improved by the overexpression of genes encoding CYPs, as well as other unidentified enzymes (Figure 7A). Genes involved in the morphophysiological traits (e.g., root development, leaf and stem development and cell differentiation) are also upregulated, which improves the plant growth to some extent under low temperatures (Figure 7B). Meanwhile, low temperatures inhibit the expression of HSPs and induce the overexpression of CRPs, which subsequently regulates the differential expression of TFs to adapt *H. perforatum* plants to low temperature conditions (Figure 7C).

## 4. Materials and Methods

### 4.1. Plant Materials

Mature seeds of *Hypericum perforatum* L. were obtained from plants cultivated in Kangxian County (33°16′20″ N, 105°31′50″ E) of Gansu Province, China in July 2016. Seedlings were induced based on a previous reported protocol [22] with slight modifications. Specifically, seeds were rinsed with tap water for 10 min and successively immersed in 70% ethanol for 2 min and 0.1% HgCl_2_ for 1 min. After each treatment, seeds were rinsed with sterile water 4 times. Sterilized seeds were inoculated on MS + 20.0 g/L sucrose + 4.0 g/L agar (pH 5.8) and germinated at 22 °C with a 24-h/d photoperiod (500 µmol·m^−2^·s^−1^ flu). After 25 days, germinated seeds were transferred to the above-mentioned basal medium (pH 5.8) supplemented with 0.5-mg/L NAA and 1.0-mg/L 6-BA for rooting and sprouting. After 20 days of growth, seedlings were treated at a constant 15 or 22 °C in a growth chamber, with each treatment containing 40 flasks (3 seedlings per flask). After 20 days of treatment, seedlings were harvested with 20 flasks used for physiological measurements and Hyp quantification and 20 flasks used for RNA-seq and gene expression.

### 4.2. Observation of Dark Gland and Leaf Stomata

The number of dark glands was calculated on a per leaf basis, with 25 leaf replicates. Stomata in the middle adaxial part of the leaf were observed by a scanning electron microscope (S-3400N, Hitachi, Tokyo, Japan), and this step was applied following a previous protocol [71].

### 4.3. Hyp HPLC Quantification

Hyp content was quantified according to previous protocols [22,72]. Air-dried seedlings were ground into powder, and a 0.1-g sample was used to extract Hyp. Ethanol extract (10 µL) was analyzed by HPLC (symmetry C18, 250 mm × 4.6 mm, 5 µm and column temperature 60 °C; Waters ACQUITY Arc, Waters Corporation, Milford, MA, USA). Hyp content was evaluated with a peak area comparison with a reference standard (hypericin, 56690; Sigma Chemical Co., St. Louis, MO, USA).

### 4.4. Total RNA Extraction and Illumina Sequencing

Total RNA samples of 15 or 22 °C with three biological replicates were extracted using a RNA kit (R6827, Omega Bio-Tek, Inc., Norcross, GA, USA) according to the manufacturer’s protocols. To ensure the quality for sequencing, the integrity of total RNA was evaluated by 1.0% agarose gel electrophoresis (Appendix A), the purity of total RNA was determined by a NanoDrop spectrophotometer 2000 (Thermo Fisher Scientific, Waltham, MA, USA), the concentration of total RNA was quantified by a Qubit2.0 Fluorometer (Thermo Scientific) and the accurate integrity of the total RNA was detected on an Agilent 2100 Bioanalyzer (Agilent Technologies Inc., Santa Clara, CA, USA). The processes of enrichment, fragmentation, reverse transcription, synthesis of the second-strand cDNA and purification of cDNA fragments was applied following the previous protocols [47]. RNA-seq was conducted using an Illumina HiSeqTM 4000 platform (Gene Denovo Biotechnology Co., Ltd., Guangzhou, China).

### 4.5. Sequence Filtration, Assembly, Unigene Expression Analysis and Basic Annotation

Raw reads were filtered according to previous descriptions [47]. Specifically, read pairs were used to filter with the principles: (1) remove reads containing adapters, (2) remove reads containing more than 10% of unknown nucleotides (N) and (3) remove low-quality reads containing more than 50% low-quality (Q-value ≤ 20) bases. De novo assembly of clean reads was performed using Trinity [73]. The expression level of each transcript was calculated and normalized to Reads Per kb per Million (RPKM) [74], and DEGs were identified according to a criteria of |log_2_(fold-change)| ≥ 1 and *p* ≤ 0.05 by DESeq2 software and the edgeR package [75,76]. Unigenes were annotated against the databases, including NR, Swiss-Prot, KEGG, KOG and GO [77].

### 4.6. qRT-PCR Validation

The primer sequence (Appendix A) was designed on the primer-blast of the NCBI website. First-strand cDNA was synthesized using a RT kit (KR116, Tiangen, China). PCR amplification was performed using a SuperReal PreMix (FP205, Tiangen, China). Melting curve was analyzed at 72 °C for 34 s. *Actin* gene (Accession no. CP002685.1) was used as a reference control. Relative expression level (REL) was calculated using the 2^−△△Ct^ method [78].

### 4.7. Statistical Analysis

All the measurements were performed using three biological replicates. A *t*-test was applied for independent samples, with *p* < 0.05 considered significant.

## 5. Conclusions

From the above observations, lower temperatures improve the plant growth and Hyp accumulation in *H. perforatum*. The DEGs observed in *H. perforatum* grown at different temperatures strongly indicate there is a transcription-based regulation of Hyp biosynthesis at lower temperatures. The master genes involved in Hyp biosynthesis, the morphophysiological traits and the stress response were identified, and their causative roles should be investigated by gene editing, like CRISPR/Cas 9 systems.

## Figures and Tables

**Figure 1 molecules-26-02294-f001:**
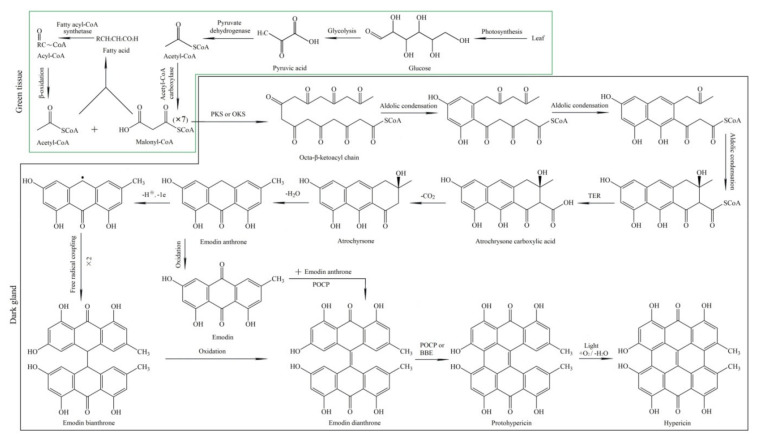
Pathway of Hyp biosynthesis from glucose to acetyl- and malonyl-coenzyme A (CoA) in the cells of green tissues (green frame) and from acetyl- and malonyl-CoA to Hyp in the cells of dark glands (black frame). PKS, polyketide synthase; OKS, octaketide synthase; TER, thioesterase; POCP, phenoloxidative coupling protein and BBE, berberine bridge enzyme.

**Figure 2 molecules-26-02294-f002:**
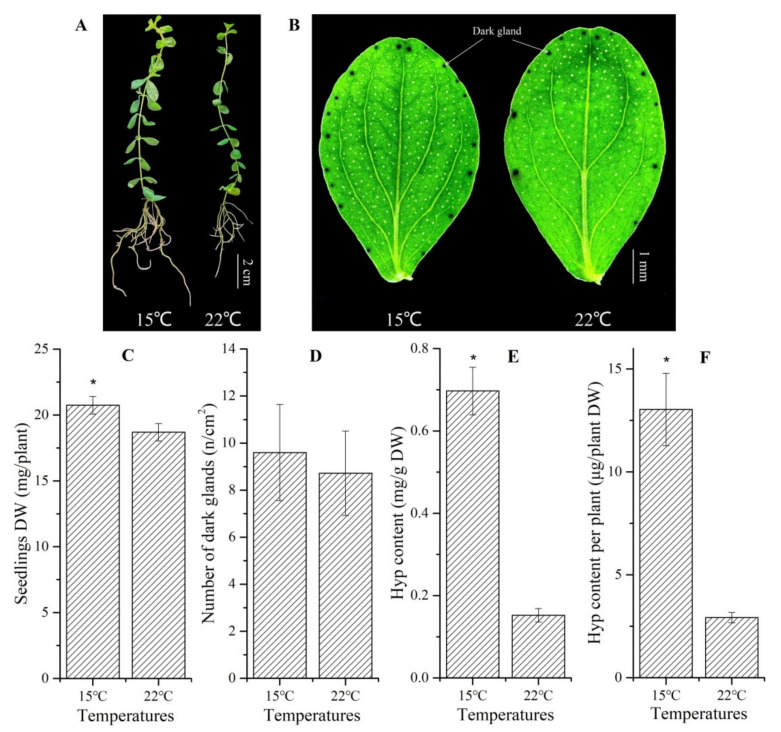
Biomass, dark glands and Hyp content in *H**ypericum perforatum* seedlings grown at 15 and 22 °C. Dry weights (DW) of whole seedlings (**A**,**C**), number of dark glands (**B**,**D**) and Hyp contents on a DW and per plant basis ((**E**,**F**), respectively). The “*” is considered significant at *p* < 0.05 between 15 and 22 °C.

**Figure 3 molecules-26-02294-f003:**
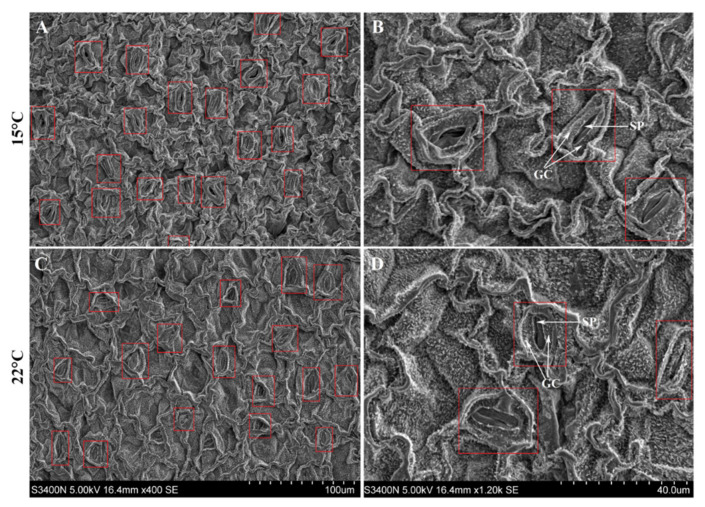
Stomata structure of *H. perforatum* seedlings grown at 15 and 22 °C. GC, guard cell and SP, stomatal pore. Images (**A**,**B**) represent the scope 100 and 40 μm at 15 °C, images (**C**,**D**) represent the scope 100 and 40 μm at 22 °C, respectively.

**Figure 4 molecules-26-02294-f004:**
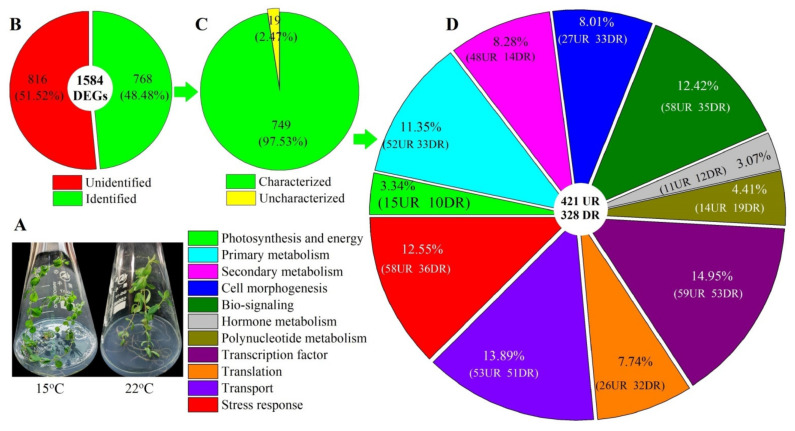
Distribution and classification of differentially expressed genes (DEGs) at 15 versus 22 °C. UR, upregulated and DR, downregulated. Image (**A**) represents the phenotype at 15 and 22 °C, image (**B**) represents the classification of unidentified and identified genes, image (**C**) represents the classification of uncharacterized and characterized genes, and image (**D**) represents the classification of the functional genes.

**Figure 5 molecules-26-02294-f005:**
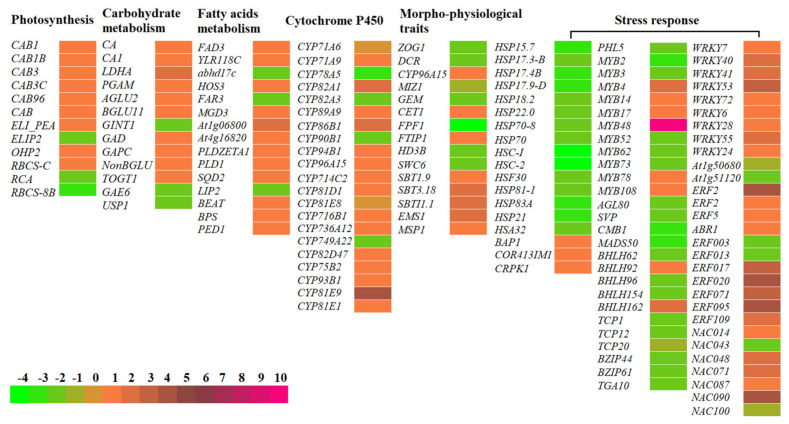
Heat map of the differential gene expressions at 15 versus 22 °C, based on the normalized Reads Per kb per Million (RPKM) values. Abbreviations of the genes are listed in Appendix A.

**Figure 6 molecules-26-02294-f006:**
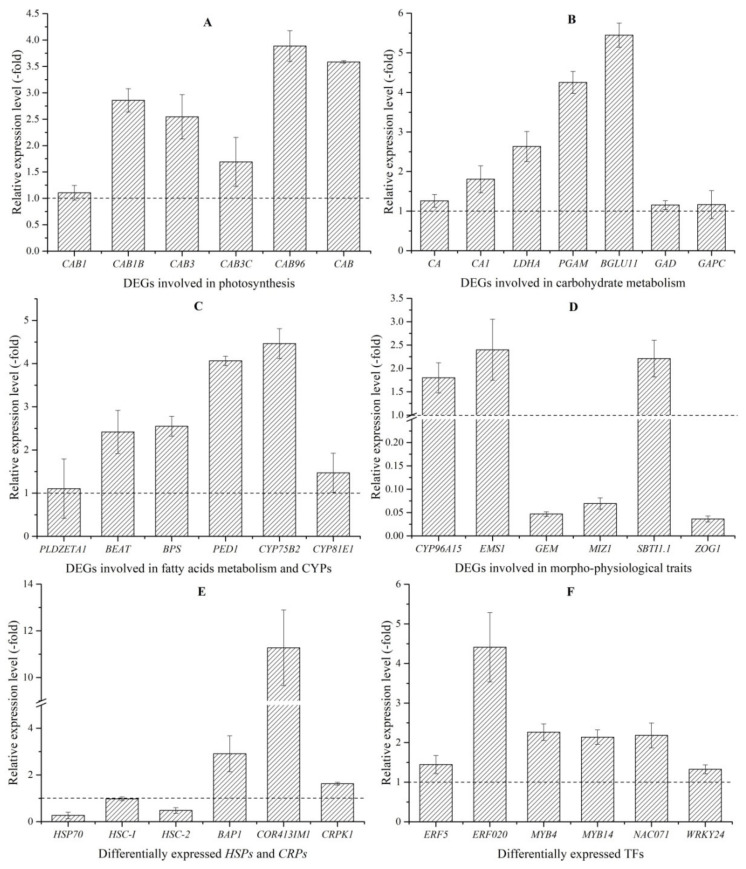
The expression level of genes involved in photosynthesis (**A**), carbohydrate metabolism (**B**), fatty acid metabolism and cytochrome P450 (CYPs) (**C**), morphophysiological traits (**D**), differentially expressed heat shock proteins (HSPs) and cold-responsive proteins (CRPs) (**E**) and transcriptions factors (TFs) (**F**) at 15 versus 22 °C plants, as determined by qRT-PCR.

**Figure 7 molecules-26-02294-f007:**
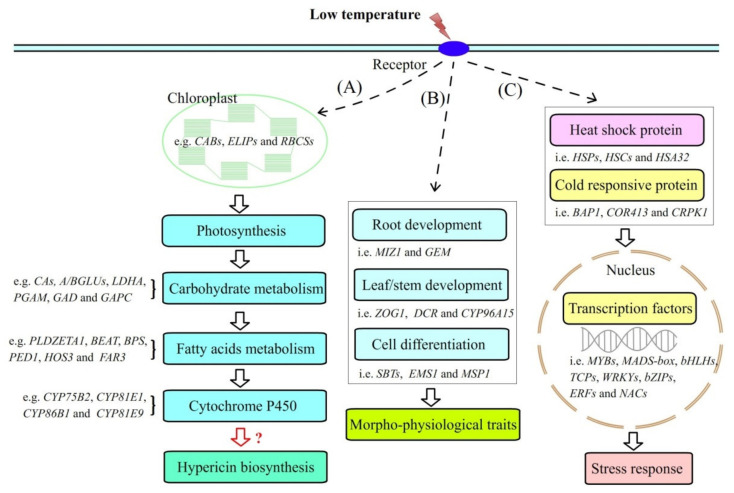
Schematic representation of the proposed pathways of the genes for regulating Hyp biosynthesis, the morphophysiological traits and the stress response in *H. perforatum*. Image (**A**) shows the genes involved in Hyp biosynthesis, image (**B**) shows the genes involved in morphophysiological traits, image (**C**) shows the genes involved in stress response.

**Table 1 molecules-26-02294-t001:** Stomata characteristics of *H**ypericum perforatum* seedlings grown at 15 and 22 °C.

Temp.	Density (mm^2^)	Opening Percentage (%)	Length (μm)	Width (μm)	Ratio of Length/Width
15 °C	22.53 ± 2.28 *	91.59 ± 0.08 *	13.27 ± 1.40	5.13 ± 1.01 *	2.69 ± 0.58 *
22 °C	19.80 ± 2.39	66.21 ± 0.05	13.42 ± 1.31	4.44 ± 1.23	3.20 ± 0.78

Values are average with their standard deviations (*n* = 30). The asterisk (*) represents a significant difference (*p* < 0.05) between 15 and 22 °C.

## Data Availability

The datasets are publicly available at NCBI, with BioSample accession: SAMN13722168 and SAMN13722169 and Sequence Read Archive (SRA) (https://dataview.ncbi.nlm.nih.gov/object/PRJNA598926) (Access: SRR10836836 to SRR1083684101, 1 February 2021).

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
