# Peer review of "Physiological and Transcriptomic Analysis Provide Insight into Low Temperature Enhancing Hypericin Biosynthesis in Hypericum perforatum"

_molecules, 2021, doi:10.3390/molecules26082294_

Round 1

Reviewer 1 Report

Comments for the authors:

The manuscript “Physiological and transcriptomic analysis provide insight into low temperature enhancing hypericin biosynthesis in Hypericum perforatum” provides a long list of DEGs that is interesting for the Hypericum scientific community. The effects of temperature on hypericin biosynthesis are of scientific and economic importance for the pharmaceutical industry. The authors report many differentially expressed genes in a very encyclopedic way. In essence, the main text is a long list of genes that the authors struggle to put together in an organic and cohesive way. This is due to the fact that the experiments reported in this study are not enough to isolate the candidate genes involved in the temperature response that determines a great variation in hypericin content. More temperatures and more genotypes should have been used. This would have allowed to reduce drastically the list of candidates and would have increased the reliability of the identified candidate genes.

Overall, the study lacks innovation and the experiments, although correctly performed, do not deliver a clear and cohesive result. Nevertheless, the study's data are valuable and could be useful for other researchers dealing with the characterization of the hypericin biosynthesis pathway.

One of the biggest flaws of the manuscript is the very low quality of the written English. It is absolutely necessary to have a deep revision of the grammar and structure of the sentences, especially in the discussion section where it seems that an automatic translation software was used.

Below are some suggestions for the improvement of the manuscript.

Line 17 and 18: it is not clear if the greater content of hypericin was detected at 15 or 22°C. Please improve the grammar and the writing style.

Line 38: The authors should include newer literature regarding the dissection of the hypericin pathway and the screening of germplasm. There are new studies (Sotak 2016, Rizzo 2019) that deal with the regulators of dark glands development in which new versions of the hypericin pathway as well as germplasm screening of Hypericum can be found.

Line 54: The authors claim that the Hyp-1 enzyme is involved in hypericin biosynthesis. Even if this is an information from older literature, it is also known in the Hypericum community that Hyp-1 is not involved in hypericin biosynthesis. This is also showed in the work from Kosuth et al 2010. Please remove this part. An updated version of the hypericin biosynthesis can be found in most recent literature, including a review published on “Genes” (another MDPI journal) by Rizzo et al on 2020.

Line 61 to 65: The role of Hyp-1 in hypericin biosynthesis has never been demonstrated. Please remove this.

Line 74: The authors must state in a clear way that no significant difference was observed in the number of foliar dark glands across different temperatures.

Line 95: Why the authors mapped only 2.5 and 2.9 million reads? Why did they not use all the reads for mapping?

Line 151: “DEGs involved in response to stress response”. Please correct this title.

Line 161: Scientific names must be written in italic.

Line 162: “DEGs involved in MYB transcription factor”. This title does not make sense. MYB TFs are not a process but a class of TFs. A correct title would be simply “Difeerentially expressed MYB transcription factors”. Nevertheless, it is not clear why the authors focus on MYB TFs. Are these the only differentially expressed TFs detected in this study? If not, why the MYBs have a dedicated section and not other TFs?

Line 180: temperature response is not one of the categories of DEGs mentioned before in the main text. Maybe the authors refer to stress response? If so, please change the text accordingly.

Line 188: Here the authors mention again a higher number of dark glands as a main result of this study but this is not true because the difference in dark glands number across temperature is not significant. Please remove this result from the discussion.

Line 190: the fact that hypericin biosynthesis depends on photosynthesis does not mean that it is involved in photosynthesis. Please correct this part.

Line 242: Please describe in detail your germination protocol.

Line 260: Please describe your RNA extraction protocol in detail. Did you perform RNA quality check with a Bioanalyzer? If yes please provide the obtained RIN factors. If another quality check method was used, please write it in this section.

Line 262: Which kind of reads was used? Single? Read pairs? Which read length was used? Please provide this information.

Line 265: De novo must be always written in italics.

Line 267: Did the authors adjust the P-values for multiple testing? If not these results are not reliable. If yes, which correction method was used? Which FDR threshold was used to call differential expression?

Line 264 to 269: Name of software or tools used are missing as well as version numbers and references for each used tool.

Line 275: Which actin gene was used as reference? Was this the only reference gene used in this study?

Line 281: Once again the author report the higher number of dark glands as a clear result of this study. This is false because no significant difference in the dark glands number was observed across temeperatures.

Reviewer 2 Report

In the present study (by Su et al.) an effort has been made to identify the genes regulating the hypericin biosynthesis. They have captured the phenotypic traits also under a different temperature gradient. The results are informative, but in the present form of MS, it lacks the proper analysis and justification of the results obtained. My comments are-

M&M section:

  1. Stomata: It is not clear from which part of the leaf is used for stomata (Mid, tip, or lower end, also abaxial or adaxial). This may significantly affect the results and observation.
  2. Total RNA extraction: As the quality of samples reflects the quality of sequencing, I miss here the details of the quality check of RNA samples.
  3. Global gene expression: This needs more elaboration to show the quality and significance of sequencing, for example, how many libraries were sequenced, if in replicates, which package was used to analyze count data, please provide the PCA plots to show the distribution pattern of the gene’s expression. Also, a correlation plot is required to make a proper conclusion.

Figures-:

  1. The representation of RNAseq data and analysis is required (see my previous comments).
  2. The heatmap scale depicted in figure 5 should indicate what it represents, FPKM values/0r normalized value of FPKM?

Results:

The expression of the DEGs identified in the study cannot always be linked with the hypericin biosynthesis, hence a correlation study between the DEGs and morpho-physiological traits under different temperatures should have been done to identify the key genes among the pool of DEGs.  This will help in the categorization of genes and linking them to desired attributes (In this case both morpho-physiological traits and hypericin biosynthesis).

Discussion:

How the DEGs identified in this study regulates low temperature-based hypericin biosynthesis is unclear. This needs a proper illustration of the mechanism both in the form of a figure and clear justification in the discussion section.

Reviewer 3 Report

The paper is read with some difficulty both for the English and for the scientific soundness. The authors aim to find genes involved in the mechanism increasing hypericin biosynthesis by low temperature exposure. In general they report DEGs that are involved in very upstream steps except for the CYPs class, some MYB TFs are also reported but related to cold response. They do not find genes involved in downstream steps such as PKSs and POCPs (Hyp-1) as reported in their recent paper Yao et al., 2019 in Plant Phys Biochem, where PKS and Hyp-1 are induced by low temperature, this should be reported and commented and most of all they should explain if PKS and Hyp-1 are not DEGs in the current experiment.

The recent papers Sotak et al 2016, Frontiers in Plant Science, Rizzo et al, 2019 Plant Biotech Journal report novel POPC genes and interestingly Rizzo et al 2019, also report a berberine bridge enzyme that could be involved in the C-C bond formation.  The authors should comment the mentioned papers and if the genes reported are not part of their DEGs. The way the paper is written makes the reader think that the downstream genes are not found, except for the class of CYPs which are not yet involved in the pathway by current literature but they could represent a novel interesting finding. CYPs genes apart, the results point to an effect of lower temperature mostly on the early steps of the pathway feeding early precursors, this should be commented and more discussed. However, I believe that based on their previous paper Yao et al., 2019 in Plant Phys Biochem, PKS and POPCs genes are also regulated by low temperature. Indeed the discussion is a general literature report of the known function of the DEGs divided by class (carbohydrate, fatty acid metabolism etc.) and it does not suggest a mechanism that the authors can suggest for the increase of hypericin under low temperature.

Concerning literature citations and the proposed hypericin biosynthetic pathway, the recent review Rizzo et al 2020 in Genes is providing an excellent up-to-date overview about genes putatively involved in the pathway and suggests a novel hypotetical pathway, it has to be commented; it is hard to believe that the authors have missed several recent papers. I think that the introduction should be re-written.

In addition the phenotypic analysis reports an increase in biomass and alteration in morphogenesis by low temperature exposure, this is not analysed in the transcriptomic part of paper at all, there is a class of DEGs related to cell morphogenesis. The authors should comment also on possible mechanisms that enhance growth at low temperature as they report the data, something is already analysed in their previous paper Yao et al. 2019 where they find enhanced expression of genes related to growth in terms of photosynthesis genes.

Some corrections:

line 49: were is not correct

line 54: And at the beginning is incorrect, yielded not yield; concerning hyp-1 gene at this point I think the paper Bais et al, 2003 JBC which reports the hyp-1 gene function should be cited.

line 68-71: the English has to be corrected, also at this point the authors should make their aim more clear which is to unravel genes involved in growth enhancement and hyp accumulation upon low temperature induction

Fig.2 legend: "the same below" is not clear, I think it refers to the asterisk; either use "The same in Table 1, but I think it is more appropriate to add a sentence in the legend of Table 1 explaining the asterisk meaning in Table 1.

Table 1 standard derivations?

Line 144 and line 162: DEGs involved in cytochrome P450 (CYPs) and DEGs involved in MYB transcription factor is incorrect, the DEGs mentioned ARE CYPs and MYB TFs, re-write.

Conclusions: re-write

281-282: The greater plant dry weight and Hyp content, more number of dark gland, stomatal density and opening percentage at lower than higher temperature present H. perforatum plants adapting to low temperature: it is unclear, correct the English.

285-286: Master genes involved in Hyp biosynthesis and temperature responsive were mined: correct the English

286-287: The causative roles of these genes in plant adaption and Hyp biosynthesis should be investigated by gene editing like CRISPR/Cas 9 systems: causative and adaption does not sound well or correct, re-write.

Round 2

Reviewer 1 Report

The manuscript is significantly improved, however, I have few more suggestions.

Line 72: Please correct to “the knowledge of the mechanism”

Line 137-138: The Hyp biosynthesis is not initiated with the biosynthesis. These are two different processes. What you can say is that photosynthesis occurs upstream of the Hyp synthesis.

Figure 6 legend: To my understanding, the horizontal line indicates the 100% expression level at 15 °C (or is it the 22°C expression level?) In any case, please specify in the legend the meaning of the horizontal line.

Line 216: Once again you mention the higher number of ark glands as a result of your study but this is false because you do not show any statistically significant difference of this parameter. Please remove this claim from your discussion.

Line 219: a verb is missing in this sentence. Probably you mean “we found 150 genes associated with….”

The manuscript will need English proofreading before publication

Reviewer 2 Report

The MS is substantially improved comparing to the previous version, all the comments raised are now addressed, hence it can be accepted for publication in the present form.